# Analysis of Trade Flows of Ornamental Citrus Fruits and Other Rutaceae in the Mediterranean Basin and Potential for *Xantomonas citri* Introduction

**Giuseppe Timpanaro \***, **Mariarita Cammarata and Arturo Urso**

Dipartimento di Agricoltura, Alimentazione e Ambiente (Di3A), University of Catania, Via S.Sofia, 100, 95123 Catania, Italy; mariar.cammarata@gmail.com (M.C.); arturo.urso@gmail.com (A.U.)

\* Correspondence: giuseppe.timpanaro@unict.it; Tel.: +39-095-758-0305

**Abstract:** In this study, we built a basic scenario for risk assessment of the introduction of *Xantomonas citri* (*X. citri*), an agent of bacterial citrus canker, through international trade activities. According to the international phytosanitary authority European Food Safety Agency (EFSA), *X. citri* is currently included in the European Union A1 list (quarantine pests not present in the area) of the European and Mediterranean Plant Protection Organization (EPPO). Therefore, at the moment, to counter the spread of *X. citri*, some pest-specific phytosanitary requirements are foreseen in the case of citrus fruit commercial activities. One possible introduction route is through some ornamental Rutaceae, which are widely cultivated in Mediterranean countries, where they are economically important and have a social impact on the employees involved and the related industries. To assess the risk of introducing *X. citri*, we distinguished the import and export territories and the type of import material, and formulated a basic hypothesis linked to the positive correlation between commercial dependence on citrus imports from countries of the Mediterranean Basin and potential risk of invasion.

**Keywords:** ornamental citrus; rutaceous plants; commercial trade; international circulation of plant material; pest risk; risk reduction options

## 1. Introduction

Citrus bacterial canker (CBC) is a disease caused by two related but taxonomically distinct bacteria: *Xanthomonas citri* pv. citri and *Xanthomonas citri* pv. aurantifolii. The former, an agent of Asian citrus fruit canker, is the most widespread worldwide. A complete taxonomy of the pest was reported by Graham et al. [1]; in the Approved Lists of Bacterial Names, "X. citri" is used to refer to any citrus fruit cancer (hyperplasia) produced by Xanthomonas, whether Asian (X. citri pv. citri) or South American (X. citri pv. aurantifolii) groups [2,3]. *X. citri* causes evident damage to the epigeal parts of plants (leaves and branches) and, in particular, on fruits, causing them to fall and/or deteriorate, preventing their economic sale. The European and Mediterranean Plant Protection Organization (EPPO) has reported its presence in many countries in Asia, the Middle East, South and Central America, Oceania, and some regions of the African continent, but not yet in Europe. Research activities have shown that commercial import and export of plants and plant parts can promote the spread of this pest, raising some concerns [4]. To confirm the presence of *X. citri*, isolating the bacterium from lesions and performing pathogenicity tests on citrus are necessary. However, isolating *X. citri* is difficult, especially from asymptomatic plants or parts of plants. The polymerase chain reaction (PCR) screening test with specific primers, available only for hosts with major economic importance, is the only reliable method for rapid analysis of suspect samples. Immunofluorescence can also be used, but no commercial antibodies have been evaluated. Monoclonal antibodies are available for

enzyme-linked immunosorbent assay (ELISA), but are mostly advised for identification of pure cultures due to low sensitivity [5]. For these reasons, *X. citri* is considered to be a quarantine organism in Europe, for which it is necessary to take preventive measures against the introduction of infected and asymptomatic plant material [6].

The protective measures against the introduction and spread of organisms harmful to plants or plant products are regulated by Directive 2000/29/EC (and subsequent amendments and additions), among which the agents of CBC are included. In particular, the directive prevents the import of many species of the genera *Citrus*, *Fortunella*, *Poncirus*, *Murraya Königii* (since subject to *Diaphorina citri* Kuwayama infection), and related hybrids (subject to contamination), except for fruits and seeds.

The risk with these imports is connected to plant material intended for planting, since *X. citri* would find environmental conditions favorable to its development. The importation of fruit is permitted due to the low probability of transferring the bacterium to a suitable host. The Mediterranean Basin is a potentially favorable area for the spread of *X. citri* because (1) the region has the best conditions for its development because citrus species are naturally present in several territories (Italy, Spain, France, Turkey, Israel, etc.), and (2) the hypothetical colonization of *X. citri* can extend the infection to commercial citrus fruits and lead to increased losses. In 2017, the citrus crop area in the region was 1,227,456 ha, with 25,714,802 tons and a production value of USD 8.6 billion (Food and Agriculture Organization Corporate Statistical Database, FAOSTAT).

Since *X. citri* is not yet present in the European Union (EU), citrus cultivation there enjoys a high degree of protection due to the institutional systems established in the various countries. These protection systems (phytosanitary authorities) act by intervening in the absence of phytosanitary documentation (passport for import/export activities), but do not prevent international trade, since, in line with World Trade Organization (WTO) agreements, plant health checks cannot become a non-tariff barrier to free trade. The EU's phytosanitary policy is defined in Article 36 of the Treaty of Rome (1956) on the free movement of goods: "goods may be excluded or reduced from circulation where there are various reasons, including the protection of human health and the preservation of plants." The EU has become an active player in the EPPO's activities. However, several species are similar to citrus, including those in the Rutaceae family, which are increasingly being used for ornamental purposes. A strict application of the protection system could have been implemented, according to a more restrictive view, on the whole Rutaceae family, including it in the annexes of the 2000/29 EU directive, but this was not completed [7,8].

In 2014, the European Food Safety Agency (EFSA) published a scientific opinion on the phytosanitary risk of introducing *X. citri* in the territories of the European Union [9]. The risk assessment was conducted within the framework of the absence of specific EU plant health legislation and the assumption that citrus-exporting countries apply measures to reduce the risk of spreading the infection and the effects on qualitative and quantitative production. The EFSA continued the analysis by identifying seven possible routes of entry, highly unlikely for fruit and very likely for all imports of plants or parts of plants for commercial or ornamental purposes intended for planting. Similar risks exist for imports with similar purposes (planting) by tourists who are not well informed about plant health risk, especially for ornamental Rutaceae and foliage fruits. The study also registered the limited awareness of amateur nursery people.

The EFSA report concluded by encouraging a ban on the import of plants or parts of plants for the purpose of planting them for commercial use as ornamental Rutaceae, even if not yet applicable to all Rutaceae. However, recent studies lack information on the susceptibility of most parts of ornamental Rutaceae to *X. citri*, the possibility of detecting latent infections, the effectiveness of the tests, and the sample size and technical feasibility of the controls. The report identifies the quarantine structures before and after entry as a possible method to contain the spread of the infection, with varying intensity depending on the size of the shipment.

The importation of potentially infectious material by passengers may be influenced by the intensity and clarity of communication and the intensity of customs controls. The EFSA estimates that potentially

0.1% of passengers carry citrus plants for planting. Custom controls should be conducted by personnel with specific experience or professional training in recognizing ornamental Rutaceae. Other possible prescriptive measures (health certification, quarantine, etc.) are not applicable in this case according to the EFSA. In the invasion process, the potential role of commercial networks, air transport connections, geographic proximity, climate similarity, the biological wealth of the country of origin, and tourist flow are all highlighted [10].

In this study, we aim to assess the risk of introducing *X. citri* in the Mediterranean Basin, considering the trade in materials that can potentially host the pathogen. Therefore, we focused on analyzing intra- and extra-Mediterranean Basin trade flows, as the intensity of the latter could affect the risk of invasion, especially if the exchanges involve nontraditional species and are not included in the list of regional phytosanitary services controls.

To answer the research question, as a first step, we started collecting and cleaning official trade statistics data [11,12]; then, these were synthesized through the use of synthetic indicators widely used in the trade literature and interpreted thanks to comparisons with public and private subjects involved in the production and the prevention and control of invasion risk.

## 2. The Scenario: Literature Review and Phytosanitary Policy in the EU

### 2.1. Literature Review: The Role of Trade in the Spread of Alien Species

Plant import, and more generally plant material import, is known as a possible avenue by which nonnative pests from specific territories are introduced. Several studies have examined various contributions to the problem, assessed the risk of invasion, and identified some possible tools for prevention and control of the spread. The analysis is complicated due to numerous implications of the problem, including biological, physiological, ecological, environmental, economic, and social aspects.

Among the various contributions, a set of properties has been established for identifying the invasion of a species and increasing frequency of alien invasions is expected due to changes in global biogeochemistry [13]. An analysis of the terminology used in 1172 studies on plant invasions provided a definition of the nomenclature currently in use, distinguishing "native" as indigenous species, "alien" as exotic or introduced species, and "invasive" as naturalized species, with "in expansion" as an extension of the range [14].

Other research has focused on experimentation and defining a model for predicting invasions, considering the ecological properties of the species and the conditions required for risk and damage, thereby threating biodiversity [15].

From an economic point of view, biological invasions represent an unwanted consequence of human activity, with real costs for society that vary according to risk and human behavior, requiring the use of economic instruments and the development of institutions and policies for prevention, monitoring, eradication, and control. As such, the control of potentially invasive species is a "public good," since society must be protected from the risks of invasive species by placing moral responsibility on the importers [16].

The invasion risk analysis of alien species is increasingly linking biology and the economy, as required by international and national policies for the management of invasive species. In these cases, specific bioeconomic models have been developed for the estimation of biological invasion through the dynamics of interaction between species (introduction, establishment, diffusion, and impact), ecological and economic systems, and management problems, considering financial consequences and the identification of alternative strategies. Prevention has been shown to be capable of generating the highest long-term net benefits [17].

Bioeconomic modeling through the use of endogenous risk theory has facilitated jointly grasping the ecological and economic aspects of production systems to improve risk assessment and the related cost–benefit value. A company can undertake various strategic options for risk management, eradication, control, and adaptation to address invasion and dissemination [18].

An evaluation of the potential economic consequences derived from the introduction and spread of harmful organisms is known as pest risk analysis (PRA), which is based on the use of different techniques such as partial budgeting, partial equilibrium analysis, input output analysis, and computable general equilibrium analysis [19]. These techniques differ from each other due to the market mechanisms considered (relationships between supply, demand, and prices), the links between agriculture and other sectors of the economy, and the ability to assess direct and indirect effects (for example, at the economic level) of the introduction of parasites [20].

According to the International Plant Protection Convention (IPPC) and the World Trade Organization Agreement on Sanitary and Phytosanitary Measures (SPS Agreement), assessments of the economic impact of an invasion are generally developed using a qualitative approach, but this approach often lacks transparency and demonstrable objectivity. A quantitative approach is needed to help improve transparency, even if specific data and models are required that can better support a decision on pest quarantine status or justify management measures [21].

Other control evaluations [22] have focused on the following:

1. The role of the least developed countries, which are not always equipped with the means or technology to manage invasive species.
2. The role of private interest groups in the design and implementation of invasive species management policies, since, in some cases, the political contributions of these groups can lead to choosing a control level that is not optimal from a social point of view [23].
3. The leadership role of some countries or areas in international cooperation and the ability to promote investments in strategies able to increase the capacity of other nations for managing invasive species problems.
4. The level of awareness of the causes and consequences of invasive species to increase the ability of governments to prevent, control, and reduce the costs of invasive species management.

Any possible avenue for spreading alien organisms is aggravated by the development of international trade in nonnative species, so a specific line of research has focused on the economic benefits connected to this trade and on the consequential risks of introducing harmful invasive species. At the international level, different countries have implemented liberal policies that can support a growing demand for non-indigenous species, which can be introduced until they prove to be problematic [24]. Screening tools for nonnative species have improved, as has their use for the purpose of possible invasive forecasts; these can be combined with impact estimates to effectively manage the trade-off between the benefits and costs of this trade. Despite the precision, these tools are often imperfect and not always suitable for supporting the decision-making process and/or not always understood as useful by policy makers [25].

The relationship between global commercial networks and large-scale distribution of non-indigenous species has been addressed with the implementation of the 10 connectivity indices, which represent the potential role of commercial networks, air transport links, geographic proximity, climatic similarity, and the wealth of the country of origin in facilitating species invasion [26]. The theory is demonstrated according to which invasion is favored by imports of live plants and/or agricultural products from countries where the focal species is present that are climatically similar to the importing country, facilitating the development potential of a predictive framework to improve risk assessment, biosecurity, and surveillance of invasions.

Finally, with regard to protection tools, one possibility is to adopt tariff-type barriers to reduce import risks, for example, by imposing a fixed or variable import tax, also useful for the establishment of funding for research, screening of imported species, education, and eradication of past invasions [27–29]. Taxation poses two types of problems: (1) acceptance by sector operators (subjects offering plants or parts of plants) and potential demand (consumers, in general) and (2) the level of taxation that is proportionate to the seriousness of the problem, the potential susceptibility, and the availability of reliable and official statistical data on trade flows by species to prohibit those potentially invasive species.

In the literature, mandatory or voluntary-type policy options have been suggested given the difficulty in ex ante estimating the possible damage, since it is necessary to have detailed information at the sector level about potential sales and costs for nurseries. In this context, some evidence emerged, such as in North America and Canada, demonstrating that establishing an annual fee for the control of potential risks and damages is possible using the consequences encountered in other, similar infestations. For example, Barbier et al. [30] indicated that it is possible to establish an annual fee for controlling the potential risk and damage derived from the importation of nonnative plants by the North American nursery industry; however, these determinations are not generalizable in time and space. The design and implementation of market control tools need adequate ecological information on the parasite, pathogen, or organism (latency, assertion of the invasion, potential damage, etc.) and on the acceptability of the market intervention by the recipients with respect to alternative forms of protection with less impact on company profits and the demand for products. Modern screening methods can quickly and cheaply identify the potential invasiveness of nonnative species for nurseries [31]; the major international competitors, before implementing market control tools, can apply a screening policy, then progress to quarantine and phytosanitary risk assessment policies, and, finally, to a selective application of annual licenses and import taxes [32].

## 2.2. Plant Health Control System in EU and Related Policies

Market globalization and climate change have significantly changed the scenario of the defense of both agricultural and forestry plants. In Europe, the alarm is very high whenever there is the spread of new unknown diseases in the area, in the face of limited interceptions at official control points [33]. In fact, the agricultural, rural, forest and landscape heritage, biodiversity, ecosystem services, and public and private green areas in the European Union, and consequently the possibility of their achieving income, employment, innovation, and food security, are at risk [34].

The phytosanitary system is regulated by international standards that arise from general agreements on the exchange of goods and services. Among them, the EU joined the IPPC, an agreement originally signed in 1952 by 182 countries within the Food and Agricultural Organization (FAO), with the aim of protecting cultivated and spontaneous plants from introduction and the spread of harmful organisms (HOs).

In this context, the FAO published the International Standards for Phytosanitary Measures (ISPM), procedures on phytosanitary measures that were revised over time to take the SPS Agreement. The IPPC also set up scientific organizations with specific phytosanitary tasks at the regional level; among these is the EPPO. The EPPO, an intergovernmental organization responsible for cooperation in plant health within the Euro-Mediterranean region, operates under the aegis of the IPPC, with the aim of cooperation and harmonization of plant protection systems.

Within the European Union, legislative tasks are assigned to the Directorate General for Health and Food Safety (SANTE), the main body of the European Commission, which issues the mandatory phytosanitary regulations for all member states. This is accompanied by the EFSA, an EU scientific and technical consultancy organization. Most of the EFSA's work is undertaken in response to requests for scientific advice from the European Commission, the European Parliament, and the EU member states. Within the latest updates, the National and Regional Phytosanitary Services connected online through a phytosanitary committee, with functions of surveillance, intervention in case of risk, and participation in the Permanent Phytosanitary Committee of Brussels.

Politically, the EU, in an attempt to not contrast free international trade and to respect the agreements signed in the WTO, issued directives which, despite being integrated and modified over time, have maintained three basic elements: (1) the general structure that regulates trade with third countries and circulation within the EU; (2) the delegation of control directly to the places of production of the plants; and (3) the plant passport.

After the first phytosanitary directive 77/93/EEC, Directive 2000/29/EC followed, consisting of annexes frequently updated by a special working group, which lists the nocive organisms under

quarantine and the phytosanitary requirements that the plants must meet in order to be marketed. To these was added Regulation 873 of 2016, which lists the EU protected areas that must be particularly protected by the introduction of quarantined nocive organisms. To be able to circulate in these areas, the propagation material must have reinforced phytosanitary requirements, certified by the protected zone (PZ) passport.

Plant products and foodstuffs from third countries (outside the EU) are checked, from a phytosanitary point of view, at authorized entry points (ports, airports, etc.), defined as border entry points (BEPs).

The monitoring of the national territory is organized on production entities (nurseries, above all), forest areas, urban areas, and naturalistic areas with visual inspection and with the aid of monitoring systems such as attraction traps. Support of the monitoring activity has been activated by the laboratories within the phytosanitary services, which also provides an agreement with research institutions (universities and other research centers). There is also a national monitoring program (co-financed by the European Union) of the HOs, which provides for a minimum number of checks (official investigations) to define the phytosanitary status and delimit the areas of national territory.

The plant protection authorities routinely perform analyses of nurseries and other production categories (e.g., citrus and potatoes) by visual checks, and randomly with samples for laboratory analysis. Nurseries that sell to professionally engaged operators (e.g., fruit growers) must be accredited as suppliers of propagating material, which at least meets the Community Agricultural Conformity (CAC) requirement. There is also a "voluntary" certification system that ensures higher phytosanitary guarantees of the material produced. The primary purpose of these periodic checks by the phytosanitary services is to ensure that requirements are maintained to issue the plant passport.

When the presence of a quarantine HO is verified, eradication or containment measures must be applied. For this reason, the operators of the supply chain (nurserymen, fruit growers, marketing firms, etc.) and all potentially involved subjects (nonprofessional nurserymen, consumers, etc.) who are invited to avoid the exchange of propagation material (scions, grafts, buds, etc.) or the purchase of plants of dubious origin, play an extremely important role.

Despite this organizational apparatus, 20 years after the issuance of Directive 2000/29/CE, various problems were unsolved, such as [5]:

- Insufficient attention paid to prevention as a result of increased imports of high-risk goods;
- The need to prioritize harmful agents at the EU level;
- The need for better tools to control the presence and natural spread of harmful organisms if they reach territories of the EU;
- The need to modernize and update the instruments concerning intra-EU movements (plant passports and protected areas);
- The need to find adequate additional resources.

Regulation (EU) 2016/2031 was thus issued, which entered into force on 14 December 2019. It establishes the rules for determining the phytosanitary risks represented by any species, strain, or biotype of pathogen, animal, or parasitic plant harmful to plants or plant products and measures to reduce these risks to an acceptable level. It is complementary to:

- Regulation (EU) 2017/625 of 15 March 2017, relating to official controls carried out by the competent authorities to verify whether professional operators are compliant with phytosanitary legislation and other official activities, which replaces Regulation (EC) 882/2004.
- Regulation (EU) 2014/652 of 15 May 2014, which establishes provisions for the management of expenses relating to the food chain, animal health and welfare, plant health, and plant reproductive material.
- Regulation (EU) 2014/1143 of the European Parliament and the Council of 22 October 2014, containing provisions aimed at preventing and managing the introduction and spread of invasive alien species, issued in the context of the EU's biodiversity strategy.

Ultimately, the regulatory effectiveness is changed, since the legislation is transformed from a directive to a regulation; the emphasis is on priority and prevention, transforming the current Annexes I and II of Directive 2000/29/EC, in which the regulated harmful organisms are listed according to their technical characteristics, regardless of their priority for the Union; the procedures by which the plant passport is issued and the systems of the protected areas are updated; and greater sharing of responsibilities with professional operators is envisaged.

## 3. Materials and Methods

The extensive trade network and the high mobility of people, even over long distances, make it very difficult to draw a precise map of the transmission from one area to another. For this reason, the methodology combines two approaches: elaboration and cartographic representation by geographic information system (GIS) and elaboration of trade dependency indices in the plant material trade. The first of these is traditionally used in pest risk assessment and aims to suggest a greater focus on import trade flows from areas at risk. The second approach, on the other hand, emphasizing trade interchange, points out that some countries of the Mediterranean Basin are much more exposed than others to the introduction of X. citri, since the costs of a trade restriction at the moment are considered by policy makers to be very relevant.

### 3.1. Data

To build the knowledge framework of the commercial flows of non-food vegetable materials, basic data were collected through official statistical sources. We referred to the United Nations (UN) Comtrade database, built by the United Nations Statistics Division (UNSD), which collects commercial data from more than 170 countries and territorial areas defined by the FAO and the Organisation for Economic Co-operation and Development (OECD). This source was then linked to the International Association of Horticultural Producers (AIPH) database, built by a union of several national associations of flower and ornamental plant producers. This source provides statistics on global production and flower and plant trade, aiming to stimulate the growing demand for ornamental trees, plants, and flowers worldwide; protect and promote the interests of the sector; act as an international hub for information and exchange of knowledge in the field; and disseminate the best practices in the production of ornamental plants.

After obtaining statistical information, we also used the European Union Notification System for Plant Health Interceptions (EUROPHYT) database. This database links the health authorities of the EU member states and Switzerland, the European Food Safety Authority, and the European Commission's Directorate General for Health and Safety, which is responsible for reporting eavesdropping (phytosanitary authorities of the EU member states and Switzerland provide data on interceptions with noncompliant expeditions through a direct link identifying the botanical species, the type of product, and the pathogen found); a timely alert system (addressed to all phytosanitary authorities of the member states and Switzerland for any suspicious interception); the creation of a database and information system (specifically structured for full access to data, trends analysis, and statistics production); and reporting (weekly, monthly, and yearly).

In addition to these databases, we directly collected data and information at the Ministry of Agricultural, Food and Forest Policies (MiPAAF) in Italy, regional phytosanitary services, and similar services at the national level in the Mediterranean.

The ultimate goal of building the statistical framework was to define the size of this trade by tracing imports from countries where *X. citri* is classified by the EPPO as present ("no details," "confirmed by survey," "widespread," "under eradication"). Thus, we collected UN Comtrade data from 59 countries (99 including federal states) for 2016–2018 from categories 805 (citrus fruits), 06 (0601, bulbs, tubers, corms, etc.), chicory plant (non-food), 0602 (live plants not elsewhere specified), 0604 (roots, cuttings, mushroom spawn), and 0604 (cut flowers, dried flowers for bouquets, etc., and foliage, etc., except flowers for ornamental purposes). Code 06, although not exclusively referring

to plants of the Rutaceae genus, since it includes all vegetable material for non-food use, is significant for defining a commercial risk dimension from countries and territorial areas where *X. citri* has been identified as being present.

The extraction of data relating to ornamental citrus only was difficult because "ornamental Rutaceae" falls under the eight-digit combined nomenclature tariff code 06022090, "Other" (Nomenclature des Activités économiques dans les Communautés Européennes, NACE).

### 3.2. Tools Representing the Risk of Invasion

Once the official statistics were collected, we used a geographic information system (GIS), which is powerful software for the representation of georeferenced information of a territory and the environment for management and analysis, spatial allocation of territorial resources, and the creation of decision support systems (DSSs). GISs have already been tested in the field of monitoring and management of the invasion of alien species [1].

A versatile aspect of GISs is the ability to superimpose layers of information, each of which describes a category (e.g., location of a specific production, import and export trade exchanges, tourism flows, meteorological trends, levels of humidity, seasonal rainfall, etc.), integrating numerical and descriptive data with geographic locations. The objective of this part of the work was, therefore, to provide global knowledge of the potential of *X. citri* invasion by trade flows, to follow its space–time evolution, to realistically simulate the level of risk and the problem on different territorial scales, to organize the statistical material available in themes that organically represent the aspects of specific interest, to evaluate the areas at greatest risk of exposure, and to model a possible containment strategy or a careful and motivated policy of managing the risk of invasion [31,33]. ArcGIS 9.3 software (Esri, Redlands, CA, USA) was used for this purpose.

### 3.3. Commercial Dependency Indices

We analyzed trade flows using the main indices in the literature. For this reason, we determined normalized trade balance and the Vollrath indices for measuring the relative commercial advantage.

The normalized trade balance (SN) is the ratio between the trade balance and the overall volume of trade:

$$\mathbf{SN} = \frac{\mathbf{X} - \mathbf{M}}{\mathbf{X} + \mathbf{M}} \times \mathbf{100} \tag{1}$$

where X represents exports and M represents imports. SN is an indicator of commercial specialization that varies between −100 (absence of exports) and 100 (absence of imports) and allows comparison of the commercial performance of aggregates of different products and of different absolute values.

Vollrath's relative commercial advantage index is used to determine whether the value of trade between two countries or regions is greater or less than expected based on their importance in world trade [35–38]. Positive values of this index indicate a relative commercial advantage of the country in the sector under examination (i.e., a competitive position), derived from the export and import flows. The index indicates the difference between the relative export advantage (RXA) index and the relative import advantage (RMP) index:

$$\text{Vollrath Index} = \text{RXA}_{ji,t} - \text{RMP}_{jn,t}$$

where *j* is the area/country; *i*, *n* are products; and *t* is the time period.

The RXA index measures sectoral competitiveness by relating the incidence in one country of a sector's exports to total exports, excluding those of the sector analyzed, and the incidence in the remaining countries (i.e., excluding the country in question) of the sector's exports to total exports, excluding those of the sector analyzed. Index values greater than 1 indicate a relative advantage of the country in the sector considered, and vice versa:

$$\text{RXA}_{ji,\,t} = \left(\frac{X_{jin,\,t}}{(X_{jn,t} - X_{jin,t})}\right) \bigg/ \left(\frac{\sum_{k,k \neq n} X_{jk,t}}{\sum_{k,k \neq n}(X_{ik,t} - X_{jik,t})}\right)$$

The RMP index is a similar indicator, calculated for imports:

$$\text{RMP}_{jn,t} = \left(\frac{M_{jin,t}}{(M_{in,t} - M_{jin,t})}\right) \bigg/ \left(\frac{\sum_{k,k \neq n} M_{jk,t}}{\sum_{k,k \neq n}(M_{ik,t} - M_{jik,t})}\right)$$

where $M_i$ and $X_i$ are imports and exports of plant material (per $n$ = UN Comtrade code 601, 602, 603, and 604) from area $j$ to area $i$, respectively; $M_j$ and $X_i$ are the total imports and exports of plant material in areas $j$ and $i$, respectively; and $t$ is the period (2015–2018).

The commercial flow was defined between the countries of the Mediterranean Basin (for each main country for which statistical data were found) and the rest of the world, with particular reference to the risk areas in which *X. citri* is classified by the EFSA as being present.

## 4. Results and Discussion

### 4.1. X. citri *World Distribution According to Official Interceptions*

The EUROPHYT indicates the presence of *X. citri* in Asia, Africa, and the American continent, and recently in Oceania (Western Australia, Fiji, Guam, Marshall Islands, Micronesia), although with diverse importance in the different environments (Table 1). According to the EPPO, *X. citri* is present in various forms (present, no details; present, widespread; present, restricted distribution; present, few occurrences), transience (transient, under eradication), or absence (absent, confirmed by survey; absent, invalid record; absent, pest eradicated).

**Table 1.** *X. citri* interceptions in main countries around the world (2006–2018) *.

| Country | No. | % |
|---|---|---|
| Bangladesh | 42 | 28.0 |
| Pakistan | 24 | 16.0 |
| China | 15 | 10.0 |
| India | 13 | 8.7 |
| Vietnam | 10 | 6.7 |
| Uruguay | 8 | 5.3 |
| Thailand | 7 | 4.7 |
| Argentina | 7 | 4.7 |
| Malaysia | 6 | 4.0 |
| Indonesia | 6 | 4.0 |
| Bolivia | 4 | 2.7 |
| Brazil | 4 | 2.7 |
| Others | 3 | 2.0 |
| Saudi Arabia | 1 | 0.7 |
| Total | 150 | 100.0 |

* Our elaboration based on the European Union Notification System for Plant Health Interceptions (EUROPHYT) database.

At the top of the ranking is Bangladesh with 42 interceptions, followed by Pakistan, China, India, and Vietnam (Figure 1).

Figure 1 shows that some countries in the world are particularly affected by the presence of *X. citri*, with interceptions in multiple areas requiring the activation of specific law enforcement measures. Figure 1 shows the relative weights of the different continents interested in *X. citri*, distributed as follows: Africa, 10.0%; South America, 15.4%; North America, 2.0%; China and East Asia, 42.0%; and India and West Asia, 30.6%.

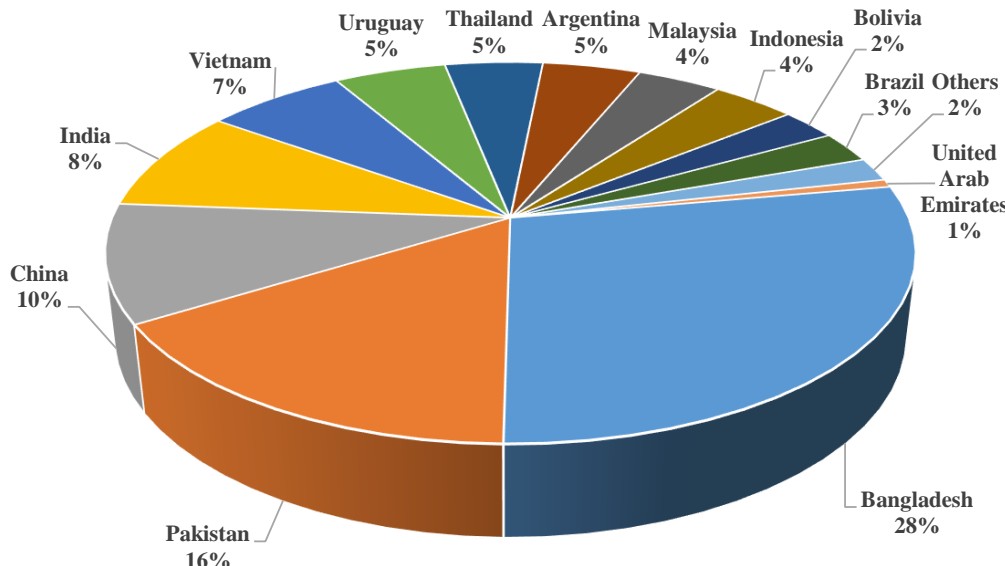

**Figure 1.** Presence of *X. citri* in the world by country in 2018.

The reports were examined for many commercial species of ornamental Rutaceae between the Mediterranean Basin and countries in North and South America, Africa, India and West Asia, and China and East Asia (Table 2). Among these, *Citrus latifolia* (around 19%), *Citrus limon* (around 13%), *Citrus aurantifolia* (12%), and Rutaceae not easily classifiable (over 19%) stand out.

**Table 2.** *X. citri* interceptions for main Rutaceae around the world (2006–2018) *.

| Species | No. | % |
|---|---|---|
| *Citrus latifolia* | 28 | 18.7 |
| *Citrus aurantifolia* | 18 | 12.0 |
| *Citrus hystrix* | 14 | 9.3 |
| *Citrus limon* | 19 | 12.7 |
| *Citrus maxima* | 13 | 8.7 |
| *Citrus reticulata* | 6 | 4.0 |
| *Citrus sinensis* | 7 | 4.7 |
| *Citrus paradisi* | 3 | 2.0 |
| *Citrus x limettoides* | 1 | 0.7 |
| *Citroncirus* | 3 | 2.0 |
| *Citrus amblycarpa* | 1 | 0.7 |
| *Citrus* sp. ** | 29 | 19.3 |
| Non-citrus species *** | 8 | 5.3 |
| Total | 150 | 100.0 |

* Our elaboration based on EUROPHYT data. ** Species not specified. *** Probable transcription mistake.

The annexes of Directive 2000/29/EC and subsequent amendments and additions have been modified over time to include some genus and related hybrids (subject to contamination), except for fruits and seeds. However, the relevant countries specialize in the production of different ornamental Rutaceae (Figure 2).

Figure 2 shows that some Rutaceae are more affected by *X. citri* than others and the varying levels in territorial areas of the world. In general, a country can build an adequate level of specialization in the production of a specific Rutaceae over time, but not show a similar ability to protect itself from invasion. Once the diffusion of *X. citri* in various areas has been ascertained, the probability of spread through trade routes becomes increasingly likely.

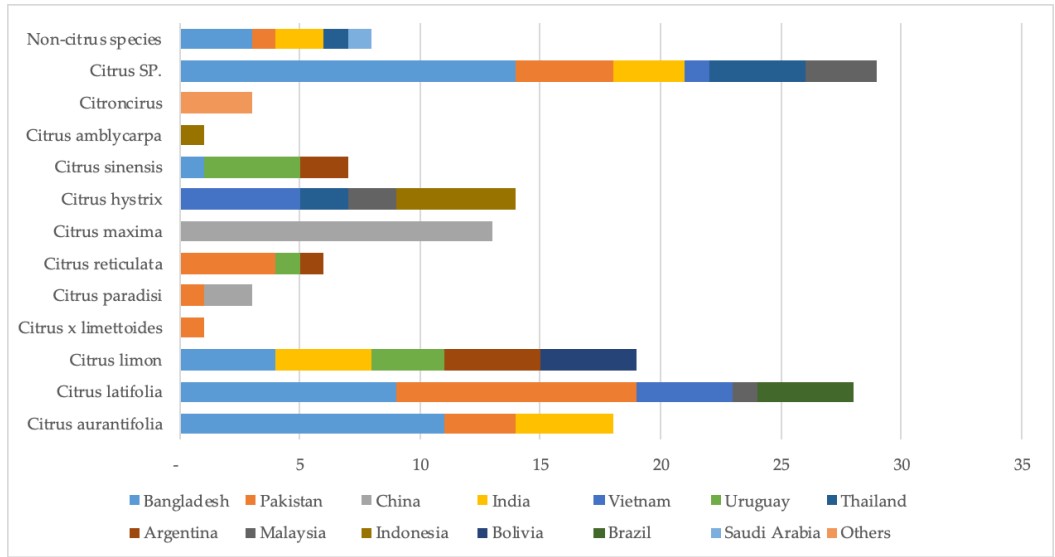

**Figure 2.** *X. citri* interceptions for main Rutaceae species around the world (2006–2018).

## 4.2. Trade in Citrus Fruits in Research Areas

History shows that plant diseases spread along trade routes; the best solution to counter this phenomenon has always been quarantine. However, this entails serious sacrifices for the economy and society, therefore policies are called upon to mediate by trying to alleviate the fallout.

**Table 3.** Import of citrus fruit in the Mediterranean Basin (2015–2018) *.

| Importing Countries | World (a) | | From Country with *X. citri* (b) | | b/a | |
|---|---|---|---|---|---|---|
| | Total Import (t) | Total Value of Import (USD) | Total Import (t) | Total Value of Import (USD) | Quantity (%) | Value (%) |
| Albania | 26,964.3 | 14,177,203.7 | 3,113.7 | 1,674,096.7 | 11.5 | 11.8 |
| Algeria | 16,551.1 | 10,625,836.7 | 953.7 | 978,204.0 | 5.8 | 9.2 |
| Bosnia Herzegovina | 64,913.5 | 28,105,727.0 | 323.6 | 367,505.0 | 0.5 | 1.3 |
| Croatia | 63,615.4 | 55,055,760.0 | 282.8 | 435,234.7 | 0.4 | 0.8 |
| Cyprus | 2488.0 | 3,759,177.3 | 588.3 | 881,329.7 | 23.6 | 23.4 |
| Egypt | 48.6 | 50,407.3 | 24.2 | 29,745.3 | 49.8 | 59.0 |
| France | 1,453,136.1 | 1,534,074,075.0 | 47,508.3 | 65,566,815.7 | 3.3 | 4.3 |
| Greece | 46,156.2 | 50,192,193.0 | 9,443.2 | 12,144,296.3 | 20.5 | 24.2 |
| Israel | 31.6 | 168,666.7 | 1.2 | 8000.0 | 3.8 | 4.7 |
| Italy | 593,871.1 | 558,081,250.7 | 41,767.9 | 55,348,820.3 | 7.0 | 9.9 |
| Lebanon | 44.9 | 56,532.0 | 1.8 | 3057.7 | 3.9 | 5.4 |
| Libya | – | – | – | – | – | – |
| Malta | 6984.1 | 6,886,291.7 | 185.4 | 185,392.0 | 2.7 | 2.7 |
| Montenegro | 10,807.1 | 9,191,797.3 | 920.0 | 1,090,121.3 | 8.5 | 11.9 |
| Morocco | 161.1 | 132,982.3 | 1.4 | 1,626.0 | 0.9 | 1.2 |
| Serbia | 114,664.2 | 72,513,896.0 | 1,896.3 | 2,352,082.3 | 1.7 | 3.2 |
| Slovenia | 69,927.9 | 59,038,815.7 | 2,591.2 | 3,936,435.7 | 3.7 | 6.7 |
| Spain | 316,028.5 | 315,008,866.3 | 95,790.6 | 111,614,613.3 | 30.3 | 35.4 |
| Syria | – | – | – | – | – | – |
| Tunisia | 82.1 | 57,033.0 | 0.2 | 74.3 | 0.2 | 0.1 |
| Turkey | 95,464.9 | 26,203,616.7 | 697.7 | 542,940.7 | 0.7 | 2.1 |
| Total | 2,881,940.5 | 2,743,380,128.3 | 206,091.4 | 257,160,391.0 | 7.2 | 9.4 |

* Our elaboration based on US Comtrade data.

Of the code 805 materials, about 2.9 million tons of citrus fruits, worth over USD 2.9 billion, are imported from the countries of the Mediterranean Basin according to UN Comtrade (Table 3). Among the countries with a large import trade flow are France (50% in volume and 56% in value of

total imports), Italy (about 21% of the volume and 20% of value), and Spain (11% in both quantity and value). As much as 7.0% of the quantity and 9.0% of the value of the total volume of imported citrus fruit are sourced from risk areas, represented by the environments in which bacterial canker is present.

The relative importance of countries changes, because when intercepting in some areas where trade values are modest (for example, Egypt), the exposure to the risk of invasion increases with the intensification of commercial flows from these areas, requiring a tightening of plant health checks. Egypt shows the highest import percentages (about 50% of the import volume and 60% of value), followed by Spain (30% and 35%, respectively), the island of Cyprus (around 24% and 23%, respectively), Greece (21% and 24%, respectively), and Montenegro (9% and 12%, respectively).

### 4.3. Commercial Flows of Import and Export of Nursery Material

For non-food vegetal materials (code 06), we found a different situation, which affects an import volume of almost 9.4 million kg with a value of over USD 46 million (Table 4).

**Table 4.** Total quantity and value of imports in Mediterranean countries of plant material for non-food uses (United Nations (UN) Comtrade code 06) from countries with *X. citri* presence.

| Importing Country | Average 2016–2018 | | | |
|---|---|---|---|---|
| | Quantity | | Value | |
| | (kg) | % | (USD) | % |
| Albania | 11.7 | 0.0 | 176,479.3 | 0.4 |
| Algeria | 24,970.7 | 0.3 | 262,013.7 | 0.6 |
| Bosnia Herzegovina | 15,757.0 | 0.2 | 95,032.3 | 0.2 |
| Croatia | 10,622.7 | 0.1 | 62,960.7 | 0.1 |
| Cyprus | 85,866.3 | 0.9 | 404,014.0 | 0.9 |
| Egypt | 81,095.0 | 0.9 | 591,665.0 | 1.3 |
| France | 1,364,510.0 | 14.5 | 7,244,000.3 | 15.7 |
| Greece | 185,893.0 | 2.0 | 1,130,429.0 | 2.5 |
| Israel | 49,538.0 | 0.5 | 334,000.0 | 0.7 |
| Italy | 3,985,943.0 | 42.5 | 21,703,219.7 | 47.1 |
| Lebanon | 202,706.3 | 2.2 | 438,338.7 | 1.0 |
| Libya | – | – | – | – |
| Malta | 1701.7 | 0.0 | 17,670.0 | 0.0 |
| Montenegro | 6934.3 | 0.1 | 24,938.3 | 0.1 |
| Morocco | 120,100.0 | 1.3 | 1,238,775.0 | 2.7 |
| Serbia | 95,116.3 | 1.0 | 135,534.7 | 0.3 |
| Slovenia | 146,386.0 | 1.6 | 343,825.0 | 0.7 |
| Spain | 1,558,099.3 | 16.6 | 8,627,688.0 | 18.7 |
| Syria | – | – | – | – |
| Tunisia | 91,928.0 | 1.0 | 397,827.0 | 0.9 |
| Turkey | 1,360,724.0 | 14.5 | 2,878,152.7 | 6.2 |
| Total | 9,387,903.3 | 100.0 | 46,106,563.3 | 100.0 |

Among the countries of the Mediterranean Basin, Italy is more exposed (with about 4 million kg and USD 21.7 million, intercepting 43% of the quantity and 47% of the value of the total imported materials), followed by Spain (about 1.6 million kg and USD 8.7 million, 17% and 19%, respectively), France (1.4 million kg and almost USD 7.2 million, 15% and 16%, respectively), and Turkey (about 1.4 million kg and almost USD 2.9 million, 15% in quantity and 6% in value).

The areas of origin are China and East Asia (5.2 million kg and USD 24.7 million), North America (nearly 1.8 million kg and over USD 10.2 million), Africa (700,000 kg and USD 4.5 million), India and West Asia (730,000 kg and over USD 4.2 million), and South America (359,000 kg and about USD 3 million), as shown in Figure 3.

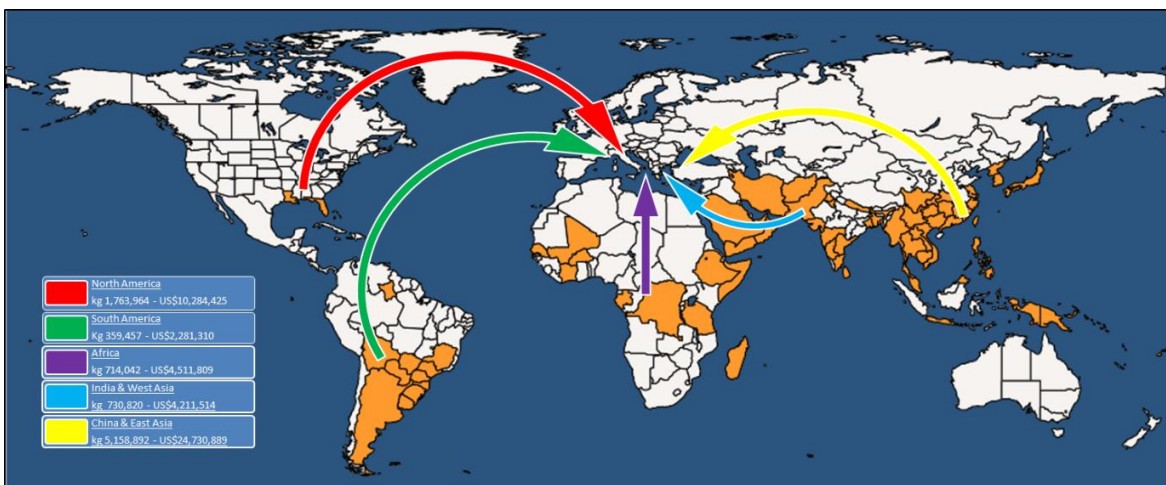

**Figure 3.** Areas from which *X. citri* may have originated in the Mediterranean Basin through imports of non-food materials in 2016–2018.

Figure 3 shows the value of imports from risk areas into the Mediterranean Basin. We built the map on the basis of EPPO analyses, to which we added average import values per area.

We repeated the same analysis with subcodes 0601, 0602, 0603, and 0604, and found that exposure to *X. citri* import risk due to international trade is limited to code 0601 (bulbs, tubers, etc.) in the countries mainly involved, and code 0602 (live plants, etc.) for Turkey (93% of imports), Italy (about 81% of imports), and Spain (59% of imported quantities), as shown in Figure 4. For Italy, a role was identified for code 0603 (cut and dried flowers, etc.), accounting for 12% of imports, whereas code 0604 (foliage, etc., except flowers) exposes Spain (12% of imports) more frequently to contamination with *X. citri*.

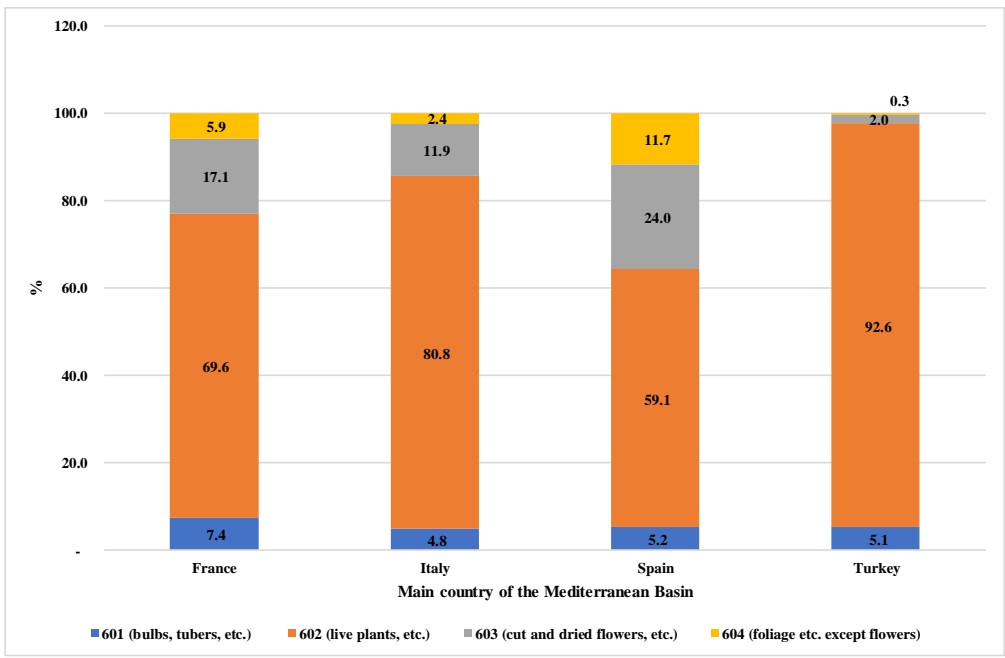

**Figure 4.** Percent of imports into Mediterranean countries of plant materials for non-food use (UN Comtrade code 06) by subcode in 2018.

Figure 4 shows that within the main producer, importer, exporter, and consumer countries of ornamental Rutaceae in the Mediterranean Basin, the importing of plant material is important.

Some countries are more exposed to *X. citri* infection because they are dependent on the importing of propagation material, in which the symptoms are not evident.

Limiting the analysis to imports from the countries at risk, we identified a partial reversal of the values for France and Italy, whereas the trends in Spain and Turkey were confirmed. Thus, in the case of France, 56% of the risk is from item 604, whereas in Italy, the risk of invasion is equally split between 602 (47.5%) and 603 (43.7%), as shown in Figure 5. However, for these countries, import from the areas with a confirmed *X. citri* problem is important.

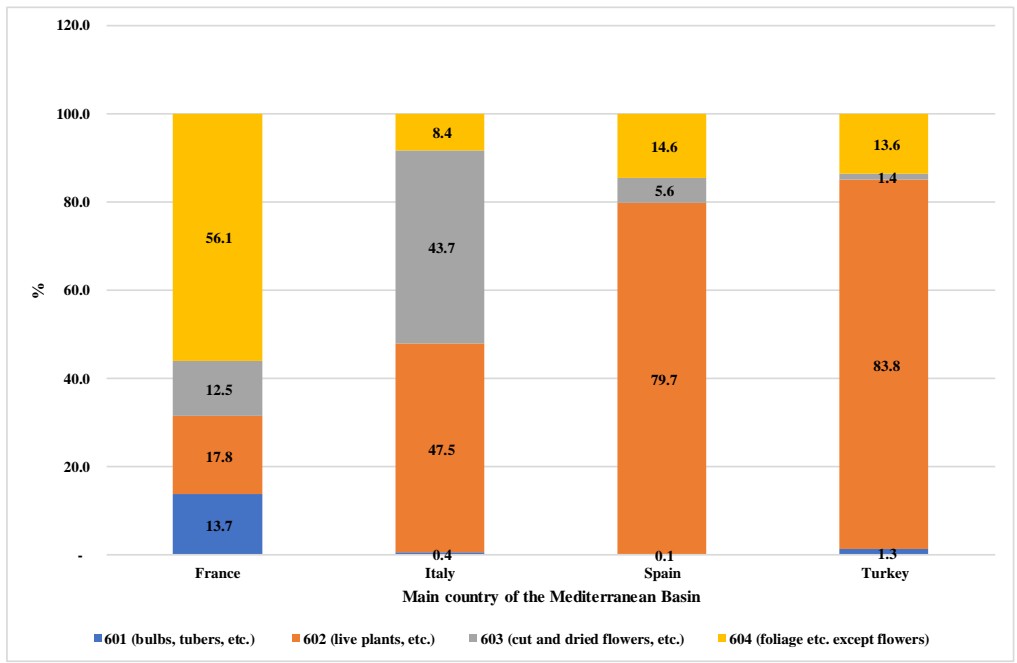

**Figure 5.** Percent of imports into Mediterranean countries of plant materials for non-food use from countries at risk (UN Trade Code 06) by subcode in 2018.

Figure 5, referring to imports from countries with a confirmed presence of *X. citri*, shows the areas at higher risk for commerce in propagating material. The characteristics of trade routes and, in particular, the departure and destination sites (physical and virtual infrastructure) and the intensity of trade can be a problem. This intensification of trade triggers an increase in demand for plant protection services with organizational problems at the institutional level. In fact, in the case of the market for "traditionally traded" species, the difficulty of controls has often resulted in a generalized extension of the ban on international trade of an ever-increasing number of ornamental Rutaceae, with a consequent reduction in benefits for consumers and profits for nurserymen. The "nontraditional" species suffer a compromise of the market potential, if they come from areas at risk.

### 4.4. Correlation between Trade and Potential Risk of Invasion

To assess the risk of potential invasion, some international trade indices were calculated to determine the competitive performance in the specific ornamental plant sector. Thus, we considered the commercial specialization for each sector as a unit for exports and imports to highlight the relative weight of the latter on the degree of production specialization.

The Vollrath indices (Table 5) indicate the different positions of specialized (positive values) and de-specialized (negative values) countries in the Mediterranean Basin. By individually observing the RXA and RMP flows, we identified a reduction in the strength of specialization, mainly due to the effect of what was observed in the various product categories. Thus, France, Italy, and Spain are mainly dependent on international trade for plant material such as bulbs, cuttings, etc. (601), as Turkey is for plants (602). For bulbs, cuttings, etc. (601), some countries bordering the Mediterranean Basin,

such as Tunisia and Egypt, have positive Vollrath index values, with a marked specialization for these materials, and corresponding negative values for live plants (602), with active and intense commercial flows like Israel and Croatia.

**Table 5.** Performance indices of the main Mediterranean Basin countries in plant material trade for non-food uses (UN Comtrade code 06, by subcode) *.

| Country | Export Advantage Index | Import Advantage Index | Vollrath Index | Country | Export Advantage Index | Import Advantage Index | Vollrath Index |
|---|---|---|---|---|---|---|---|
| | | 601 | | | | 602 | |
| Italy | 15.8 | 108.6 | −92.8 | Italy | 161.8 | 87.8 | 74.0 |
| France | 109.4 | 123.4 | −14.0 | France | 163.1 | 81.0 | 82.1 |
| Turkey | 32.1 | 93.1 | −61.0 | Turkey | 49.7 | 519.8 | −470.1 |
| Israel | 138.3 | 470.5 | −332.2 | Israel | 12.2 | 122.3 | −110.1 |
| Slovenia | 66.7 | 84.9 | −18.2 | Slovenia | 1042.3 | 128.4 | 913.9 |
| Tunisia | 190.6 | 27.1 | 163.6 | Tunisia | 2.0 | 2559.7 | −2557.7 |
| Egypt | 18,581.1 | 251.7 | 18,329.5 | Egypt | 0.6 | 282.6 | −282.0 |
| Spain | 22.6 | 77.2 | −54.5 | Spain | 240.1 | 124.2 | 115.9 |
| Serbia | 9.4 | 73.2 | −63.7 | Serbia | 758.9 | 229.9 | 529.0 |
| Croatia | 1.9 | 59.1 | −57.1 | Croatia | 80.8 | 178.3 | −97.5 |
| Greece | | | 0.0 | Greece | 2246.6 | 169.6 | 2077.0 |
| Algeria | | | 0.0 | Algeria | | | 0.0 |
| Bosnia | | | 0.0 | Bosnia | | | 0.0 |
| Cyprus | | | 0.0 | Cyprus | | | 0.0 |
| Montenegro | | | 0.0 | Montenegro | 6.3 | 250.1 | −243.7 |
| | | 603 | | | | 604 | |
| Italy | 51.1 | 109.9 | −58.8 | Italy | 180.5 | 113.7 | 66.8 |
| France | 72.6 | 115.3 | −42.7 | France | 38.4 | 112.9 | −74.4 |
| Turkey | 312.4 | 9.6 | 302.7 | Turkey | 90.3 | 20.0 | 70.4 |
| Israel | 661.7 | 17.6 | 644.1 | Israel | 265.1 | 14.8 | 250.3 |
| Slovenia | 0.7 | 78.8 | −78.2 | Slovenia | 6.0 | 100.7 | −94.8 |
| Tunisia | | | 0.0 | Tunisia | 6791.7 | 8.9 | 6782.8 |
| Egypt | 137.5 | 4.4 | 133.1 | Egypt | | | 0.0 |
| Spain | 65.5 | 147.1 | −81.5 | Spain | 35.3 | 113.2 | −78.0 |
| Serbia | 12.2 | 46.4 | −34.2 | Serbia | 25.0 | 34.5 | −9.5 |
| Croatia | 214.5 | 61.7 | 152.8 | Croatia | 59.2 | 82.0 | −22.9 |
| Greece | 10.2 | 87.2 | −77.0 | Greece | 0.0 | 96.8 | −96.8 |
| Algeria | | | 0.0 | Algeria | | | 0.0 |
| Bosnia | | | 0.0 | Bosnia | | | 0.0 |
| Cyprus | | | 0.0 | Cyprus | | | 0.0 |
| Montenegro | 4.4 | 63.1 | −58.7 | Montenegro | 5827.7 | 41.4 | 5786.2 |

\* Our elaboration based on UN Comtrade data.

We analyzed the different areas around the world using GIS cartography of the import volumes of plant material to depict the phytosanitary problem (Figure 6).

The GIS cartography in Figure 6 uses different color shades for import areas with the highest and least risk. The colors show that controls on phytosanitary services must be intensified in the south of the Mediterranean Basin, as these areas are close to countries with reports of *X. citri* presence.

The normalized trade balance further emphasizes the degree of dependence on countries with *X. citri* risk, resulting in potentially dangerous import and export movements (Table 6). Table 6 shows the commercial interchange between the main EU and Mediterranean countries and the rest of the world and areas at risk. Overall, a diversified scenario emerges, with an underlying trend common to all subsectors with a reversal of values between the normalized trade balance of each country toward the rest of the world and the normalized trade balance with respect to the risk areas. The negative values due to the prevalence of imports indicate a marked dependence on the areas at risk of Turkey for

code 601, of Italy and Spain for code 602, of Spain and Turkey for code 603, and of France and Turkey for code 604, indicating that commercial transactions of this plant material are at risk of spreading infection.

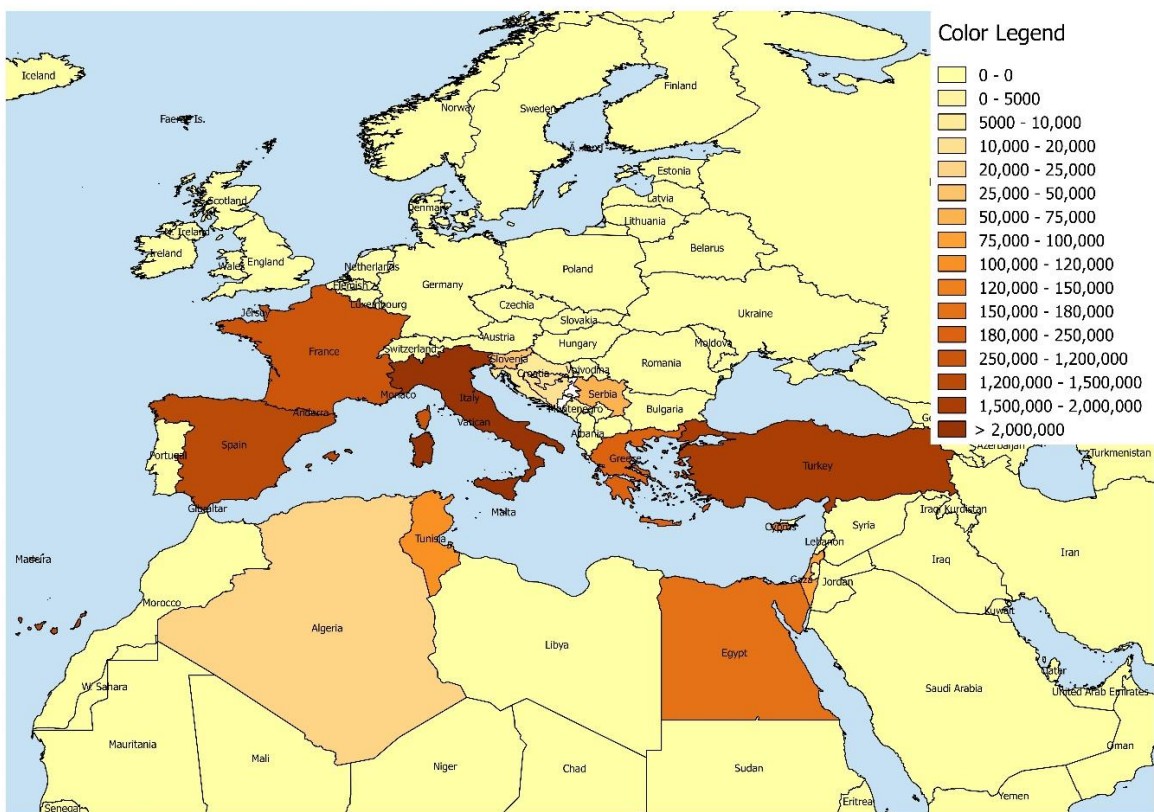

**Figure 6.** Plant material import scale (UN Comtrade code 06) from *X. citri* risk countries (2018).

**Table 6.** Trade indices in the countries of the Mediterranean Basin *.

| Country | Import | Export | Total Normalized Balance | Normalized Balance from Countries at Risk | Country | Import | Export | Total Normalized Balance | Normalized Balance from Countries at Risk |
|---|---|---|---|---|---|---|---|---|---|
| | t | t | % | % | | t | t | % | % |
| | | | 601 | | | | | 602 | |
| France | 25,717.0 | 5837.6 | −63.0 | 10.9 | France | 243,322.3 | 62,540.6 | −59.1 | 26.6 |
| Italy | 10,789.0 | 4695.0 | −39.4 | 82.7 | Italy | 183,257.6 | 462,822.8 | 43.3 | −50.4 |
| Spain | 4264.6 | 3650.6 | −7.8 | 46.6 | Spain | 48,673.3 | 193,220.3 | 59.8 | −3.1 |
| Turkey | 2887.6 | 240.4 | −84.6 | −98.2 | Turkey | 52,122.6 | 28,615.4 | −29.1 | 35.4 |
| Other (**) | 131,993.6 | 30,411.2 | −62.5 | 98.3 | Other (**) | 4,968,928.4 | 1,808,356.1 | −46.6 | 46.1 |
| | | | 603 | | | | | 604 | |
| France | 59,800.1 | 2077.4 | −93.3 | 33.2 | France | 20,563.7 | 1373.5 | −87.5 | −69 |
| Italy | 27,079.0 | 11,151.9 | −41.7 | 78.3 | Italy | 5552.6 | 21,814.7 | 59.4 | 47.6 |
| Spain | 19,758.2 | 34,780.5 | 27.5 | −46.6 | Spain | 9645.4 | 13,778.3 | 17.6 | 63.6 |
| Turkey | 1105.4 | 11,611.2 | 82.6 | −8.1 | Turkey | 176.4 | 3288.1 | 89.8 | −45.6 |
| Other (**) | 27,461.7 | 18,843.1 | −18.6 | 90.2 | Other (**) | 15,546.1 | 14,852.7 | −2.3 | 56.4 |

\* Our elaboration based on UN Comtrade data. ** Other countries of the Mediterranean Basin.

Considering the location of Italy, France, and Spain in the horticultural market, the contribution of plant material movements between these countries and China and Thailand, which are areas at risk, should be considered to identify whether the territorial areas under examination show commercial competitiveness, a precursor to a possible invasion (Figure 7).

Italy expresses a specialization toward China for the first three subsectors (601, 602, and 603) and a de-specialization for the last (604).

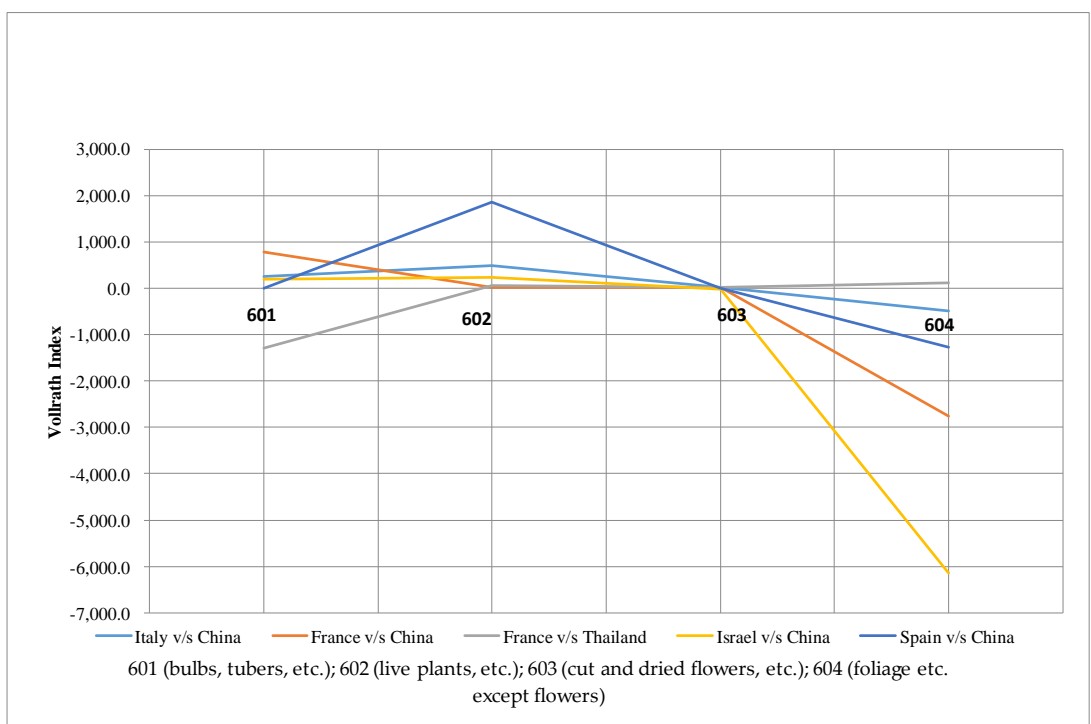

**Figure 7.** Trade between some countries of the Mediterranean Basin and some areas at risk (2018).

Figure 7 indicates the Vollrath index of trade dependence in direct exchanges between countries of the Mediterranean Basin and some countries at risk where *X. citri* is present. Spain depends on China for bulbs and cuttings, cut flowers, and foliage (all with negative values). France shows behavior similar to Italy toward China, but less toward Thailand, upon which it depends for a consistent flow of imported bulbs and cuttings.

Even with the limited data available from the statistical sources, which prevented extending the evaluation to other countries, the direct interchange shows values that are affected by economic situations, even in terms of location and assuming values in decline compared with the entity of the overall commercial transactions. The direct interchange is crucial for determining the trade balance in comparison with other countries.

The nodal point remains the qualification of nursery production according to standards that guarantee and protect quality. Certification of the origin of nursery materials represents the only way to guarantee efficiency and fairness, fundamental functions of the equation of well-being. In the countries of the Mediterranean Basin there is also a diversity of situations in terms of partnerships with universities and phytosanitary centers, which can promote the training of operators and related certification of skills, for which the initiative is often exclusively confined to individual operators and their ethical behavior.

## 5. Conclusions

The risk of *X. citri* derived from trade is linked to the type of business (distinguishing between professional operators and small producers), the type of ornamental Rutaceae (common or niche species), and the type of trade (traditional or modern and/or electronic).

There are two types of companies: professional structures and small producers. Professional structures are legally recognized by phytosanitary services and are inscribed in the regional register, authorized to issue extra-EU passports, and also collaborate with phytosanitary services for issuance of the intra-EU phytosanitary certificate. On the contrary, small producers are exempt from previous obligations, so they may be tempted to widen their commercial offerings through imports at risk.

The professional structure can also be tempted to relocate production activity to achieve greater economies of scale and scope (e.g., France in the Réunion Islands).

We emphasize the different types of Rutaceae species affected by trade. Some common ornamental Rutaceae (e.g., bitter orange, lemon, *Poncirus trifoliata*, *Citrus limonimedica*, kumquat or *Fortunella margarita*, *Citrus myrtifolia*, etc.) have limited market value that do not justify long-distance transport. However, the niche ornamental Rutaceae (e.g., *Murraya paniculata*, *Citrus mitis*, *Coleonema pulchrum*, *Poncirus trifoliata*, *Zanthoxylum beecheyanum*, *Murraya exotica*) are valuable market products, especially for hobbyists interested in commercial assortments.

Finally, it is possible to distinguish trades into traditional and modern forms, represented by multichannel, electronic, and direct distribution formulas for final consumption [39]. To date, there are two main elements for defense against invasion by dangerous pathogens: traceability and control of the supply chain at a physical location. This second chance is getting less and less common today. Indeed, the consumer can directly connect to intermediation platforms, which offer all kinds of products, including those of foreign origin either in the community or non-EU.

Then, a problem arises between trade in plants and in propagation material. The transport of plants ready for sale from farther distances creates risks, given the unlikelihood of survival of specimens 24 to 48 months of age in containers during long-distance transport. On average, transfer from far-away countries can take two to three weeks, unless high-cost transport is used, which is not economically viable since all the appropriate cultivation conditions are present in the Mediterranean Basin. So, it is possible to buy fresh plants not subjected to the stress of long transportation.

The risk of *X. citri* spread is mainly linked to propagation material (marzes, in particular) in which *X. citri* is easily concealable (in case of intentional invasion). New regulations on phytosanitary surveillance (Reg. 2031/2016) were implemented starting from 14 December 2019, which, among other things, provide for the possibility for professional operators to equip themselves with a "risk management plan related to harmful organisms"; plans that must be approved by a competent authority. The new legislation, however, appears to be strict with regard to the regulations for professional operators (for example, without compromising the total traceability extended to the growing substrates and the obligation to intervene). However, to support the free market, the regulation maintains exceptions to products (parts of plants not intended for trade) and non-professional operators, increasing the risk of possible *X. citri* invasion. These are privileges that hobbyists do not want to sacrifice.

On the subject of EU intervention, considering the numerous attempts to unwittingly and/or illegally introduce citrus and related products, which are intercepted at various airport control points, the EU issued a specific regulation, (EU) 2019/2122 on 10 October 2019, to intensify checks on passenger baggage. In particular, the new phytosanitary regulation considers the risk related to plant and country of origin. Passengers from countries where CBC is present may unwittingly introduce the bacterium and/or carriers with Rutaceae species they habitually use as spices in their diet. An example is the discovery of *Murraya Königii* (curry leaf) in the luggage of passengers from Asia (Bangladesh, Mauritius, Sri Lanka) heading to Italy.

In conclusion, in finding that the current phytosanitary control system is rigid and reliable enough to ensure protection from invasion, some initiatives to contain the spread of *X. citri* include the following:

(1)  Passengers/unintentional importation: Information dissemination needs to be increased to result in cultural change, the transport of plant material in baggage declaration should be provided at customs, routes at risk should be defined and a dedicated control system should be implemented, and the willingness of the consumer to pay a premium price for a certified ornamental product should be investigated.

(2)  Institutional intervention: Train operators and provide information, define strict control procedures for e-commerce, increase research resources, review exemptions for small producers

because EU Regulation 2016/2031 allows member states (e.g., by committing them to product traceability/voluntary certification).

The coexistence of other types of plant diseases in the same areas should be considered (for example, Huanglongbing or HLB, the agent of citrus greening, a destructive disease associated with bacteria of the genus *Candidatus liberibacter* transmitted by psyllids), along with the opportunities and risks of modern e-commerce. *X. citri* and HLB coexist in 62% of the countries in which they are found. Since both bacteria and their vectors have been included by the EPPO on the list of quarantine organisms, which would be a complementary feature of the phytosanitary passport, this would indirectly help with protection from *X. citri*.

Currently, phytosanitary defense is based on two conceptual elements: traceability and supply chain control at a physical location. Electronic commerce is now part of the supply chain of ornamental plants at different levels; sometimes, due to e-commerce, the plant or part of the plant (graft materials, for example) skips some important physical health checks. The consumer can connect directly with intermediation platforms instead of the producer platforms, which offer products of all kinds, even of foreign, community, or non-EU origin, despite EU Regulation 625/2017 on official controls. If controls are not implemented on a cross-border level, they should be implemented in situ [40–42].

Future research developments will be aimed at evaluating the possibility of using market regulation tools (for example, voluntary standards or taxation) to ensure that the nursery system supports the expected social cost of an accidental *X. citri* invasion in the Mediterranean Basin.

**Author Contributions:** The study is the result of full collaboration and therefore all authors accept full responsibility. G.T. wrote Section 1, Section 2.1, Section 3.3, Section 4.1, Section 4.4, and Section 5; M.C. wrote Sections 3.1 and 4; A.U. wrote Section 2.2, Section 3.2, and Section 4.3; M.C. processed the US Comtrade data and summarized the results in the tables. All authors have read and agreed to the published version of the manuscript.

**Funding:** This study was conducted with the financial support of the ORPRAMed Project on "Risk assessment of introduction of *Xanthomonas citri* subsp. citri through commercial trade of ornamental rutaceous plants in the Mediterranean basin", ERA-NET ARIMNET 2-Call 2015 (dott.ssa Paola Caruso, Consiglio per la Ricerca in agricoltura e l'analisi dell'Economia Agraria–CREA, scientific coordinator).

**Acknowledgments:** The authors thank the anonymous reviewers for their valuable suggestions for the improvement of the paper.

**Conflicts of Interest:** The authors declare no conflicts of interest.

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
