# Peer review of "Analysis of Trade Flows of Ornamental Citrus Fruits and Other Rutaceae in the Mediterranean Basin and Potential for Xantomonas citri Introduction"

_agriculture, doi:10.3390/agriculture10050171_

Round 1

Reviewer 1 Report

Thank you very much for answering all my questions and making alterations to your text according to my comments.

Reviewer 2 Report

The authors followed all my suggestions.

Author Response

This manuscript is a resubmission of an earlier submission. The following is a list of the peer review reports and author responses from that submission.

Round 1

Reviewer 1 Report

Overall, the paper is well written and comprehensive. The topic is interesting, since Xcc is a quarantine pathogen for many countries and especially for the Mediterranean basin but on my personal opinion it suffers from interesting forecasts for the near future. In other words, your study is very theoretical and does not provide useful information for the stakeholders. However, due to the fact that Xcc is a major threat for the Mediterranean basin, any information and any analysis could contribute towards the built up of appropriate strategies for dealing with this pathogen. Therefore, I suggest the publication of your research after minor corrections. Please see the comments bellow:

Comments

General

I believe that it would have been useful to give some information on the biology of the and the ecology of the pathogen. You don’t give any information on tests for spotting the disease. Phytosanitary checks are really important for hampering the disease entrance in other countries. Could you please add some information on this topic and suggest the most appropriate methods for screening the imported plant material and plant products? In lines 45-54 you briefly mention the strategy/policy but you do not actually give any information on the methods and the procedures that should be followed. The particular disease is difficult to spot. You may have infected plant material that do not show any disease symptoms. Have you taken this into account in your analysis for the risk of invasion? Combination of quantitative and qualitative approaches may be the more accurate methods for such analysis? Is there any common EU policy for this pathogen? If yes please provide some information on this policy. If not, please discuss the dangers of not affiliating such policy. It would have been useful to provide a forecast on the risk of invasion for each EU country based on the data that you analysed.

Author Response

(Reviewer 1): Comments and Suggestions for Authors:

Overall, the paper is well written and comprehensive. The topic is interesting, since Xcc is a quarantine pathogen for many countries and especially for the Mediterranean basin but on my personal opinion it suffers from interesting forecasts for the near future. In other words, your study is very theoretical and does not provide useful information for the stakeholders. However, due to the fact that Xcc is a major threat for the Mediterranean basin, any information and any analysis could contribute towards the built up of appropriate strategies for dealing with this pathogen. Therefore, I suggest the publication of your research after minor corrections. Please see the comments bellow:

R: Many thanks for what you said about the paper

Comments General

I believe that it would have been useful to give some information on the biology of the and the ecology of the pathogen.

R: I have included in Introduction further clarification and a new bibliographical reference

You don’t give any information on tests for spotting the disease. Phytosanitary checks are really important for hampering the disease entrance in other countries. Could you please add some information on this topic and suggest the most appropriate methods for screening the imported plant material and plant products?

R: In Introduction I dealt with this problem. EPPO reports a practice of control tests, but these are only effective to the extent that the disease is evident. It is no coincidence that the European Union has issued Regulation (EU) 2016/2031 of 26 October 2016 (which came into force on 31 December 2019) and, with Regulation (EU) 2019/2122 of 10 October 2019, has required that passenger baggage checks be intensified, especially for passengers from countries at risk. You will also find additions in Conclusion

In lines 45-54 you briefly mention the strategy/policy but you do not actually give any information on the methods and the procedures that should be followed. The particular disease is difficult to spot. You may have infected plant material that do not show any disease symptoms. Have you taken this into account in your analysis for the risk of invasion? Combination of quantitative and qualitative approaches may be the more accurate methods for such analysis? Is there any common EU policy for this pathogen? If yes please provide some information on this policy. If not, please discuss the dangers of not affiliating such policy. It would have been useful to provide a forecast on the risk of invasion for each EU country based on the data that you analysed.

R: EPPO and EFSA are the two institutional reference actors in the organisation of the plant health control system at European level. In particular, EPPO (European and Mediterranean Plant Protection Organization) was established in 1951 at the end of a process started in 1903 with the establishment of an international "Standing Phytopathological Committee", created with the task of preventing the spread of epidemic diseases through the control of plant material in trade relations. Similar organisms were born in other parts of the world, with the aim of not hindering the free movement of plant materials except in cases of obvious possible damage to human or plant health. For Xcc, included in EPPO list A1, it is foreseen that citrus-growing countries should prohibit import of plants for planting (except seeds and tissue cultures) of Rutaceae from countries where X. axonopodis pv. citri occurs. They may also prohibit import of rutaceous fruits from the same source.

I have added a clarification note

R: I asked the journal service for a language review, as suggested.

Thank you for your valuable suggestions.

Reviewer 2 Report

The paper “Analysis of trade flows of ornamental citrus fruits and other rutaceae in the Mediterranean basin and potential for introduction of Xantomonas citri pv. citri” has been reviewed and the report is included below.

The manuscript Agriculture-610743 reports a risk assessment of the introduction of Xantomonas citri, a random agent of Bacterial Citrus Cancer, through international trade activities. The topic is of high relevance for agriculture management and the approach is adequate. Although the study is innovative and unique, unfortunately, the manuscript, in the present state, is not suitable for publication due to some lacks. Therefore, to be accepted for publication, the following minor revisions are required.

My main concerns are about the introduction section. Authors often refer to specific concepts without reporting any references. Following some examples:

Line 42-44. “The risk is, in fact, connected to the importation of plant material intended for planting, since Xcc would find environmental conditions favorable to its development, but not from the importation of fruit due to the low probability of transferring the bacterium to a suitable host.”

Line 55-56. In 2014 the EFSA (European Food Safety Authority) published a “scientific opinion” on the phytosanitary risk of introduction of Xcc on the territory of the European Union.

Moreover, authors should improve the English style overall the manuscript using a specific English revision service.

Author Response

Comments and Suggestions for Authors

The paper “Analysis of trade flows of ornamental citrus fruits and other rutaceae in the Mediterranean basin and potential for introduction of Xantomonas citri pv. citri” has been reviewed and the report is included below.

The manuscript Agriculture-610743 reports a risk assessment of the introduction of Xantomonas citri, a random agent of Bacterial Citrus Cancer, through international trade activities. The topic is of high relevance for agriculture management and the approach is adequate. Although the study is innovative and unique, unfortunately, the manuscript, in the present state, is not suitable for publication due to some lacks. Therefore, to be accepted for publication, the following minor revisions are required.

My main concerns are about the introduction section. Authors often refer to specific concepts without reporting any references. Following some examples:

Line 42-44. “The risk is, in fact, connected to the importation of plant material intended for planting, since Xcc would find environmental conditions favorable to its development, but not from the importation of fruit due to the low probability of transferring the bacterium to a suitable host.”

R: The Introduction is the part where a lot of work has been done. New literature references have been introduced and aspects of biology and potential harm have been clarified. EPPO suggests the prevention instrument to be adopted. For Xcc, included in EPPO list A1, it is foreseen that citrus-growing countries should prohibit import of plants for planting (except seeds and tissue cultures) of Rutaceae from countries where X. axonopodis pv. citri occurs. They may also prohibit import of rutaceous fruits from the same source.

Line 55-56. In 2014 the EFSA (European Food Safety Authority) published a “scientific opinion” on the phytosanitary risk of introduction of Xcc on the territory of the European Union.

Moreover, authors should improve the English style overall the manuscript using a specific English revision service.

R: I asked the journal service for a language review, as suggested.

Thank you for your valuable suggestions.
